# Attitude, preparedness, and perceived self-efficacy in controlling COVID-19 pandemics and associated factors among university students during school reopening

**Mesfin Tadese** *, **Abebe Mihretie**

Department of Midwifery, College of Health Sciences, Debre Berhan University, Debre Berhan, Ethiopia

* mesitad031@gmail.com

## Abstract

### Introduction

The coronavirus disease 2019 (COVID-19) pandemic remains a significant public health problem globally. In Ethiopia, the number of infected peoples and deaths due to COVID-19 has increased dramatically in the past. Currently, students are resuming to face to face education with strict prevention measures. University students are more dynamic and more susceptible to acquiring and spreading the virus.

### Objective

To assess the attitude, preparedness, and self-efficacy to prevent and control COVID-19 and associated factors among university students during school reopening, Northeast Ethiopia.

### Method

A cross-sectional study was conducted among Debre Berhan University (DBU) students from December 1 to 15/2020, when students return to campus. A multistage sampling technique was applied to recruit 682 participants. The ReadyScore criteria were used to classify the level of preparedness. Epi-Data version 4.6 was used for data entry, while SPSS version 25 for analysis. Descriptive and Binary logistic regression analysis was computed, and a *p*-value < 0.05 was considered statistically significant.

### Result

The overall level of favourable attitude, good preparedness, and high self-efficacy among students were 67.2%, 17.9%, and 50.4%, respectively. Only mothers' education was associated with attitude. Female gender, open relationships, health science faculty, heart disease, and favourable attitude were significant preparedness factors. Whereas being undergraduate, parents' education, residing in dorm being four and above, having kidney disease, having friend/family history of COVID-19 infection and death, favourable attitude, and good preparedness were predictors of self-efficacy.

**Data Availability Statement:** All relevant data are within the paper and its Supporting Information files.

**Funding:** The authors received no specific funding for this work.

**Competing interests:** The authors have declared that no competing interests exist.

**Abbreviations:** AOR, Adjusted Odds Ratio; CI, Confidence Interval; COR, Cruds Odds Ratio; COVID-19, Coronavirus Disease 2019; DBU, Debre Berhan University; SPSS, Statistical Packages for Social Sciences; SRS, Simple Random Sampling; US, United States; WHO, World Health Organization.

## Conclusion

The level of attitude, preparedness, and self-efficacy towards COVID-19 among students during campus re-entry were low. Managing chronic illnesses and raising the attitude and preparedness of students is essential to reduce the burden of COVID-19 pandemics. Besides, emphasis should be placed on male, unmarried, postgraduate, and non-health science students to increase the level of preparedness and self-efficacy.

## Introduction

The novel coronavirus disease 2019 (COVID-19), an infection caused by a severe acute respiratory syndrome coronavirus 2 (SARS-CoV-2), remains a global public health crisis. In December 2019, the virus was first isolated and identified from clusters of patients with pneumonia of unknown origin exposed at a seafood wholesale market in Wuhan City, China. It is the seventh member of the family of coronavirus that infects humans [1]. On January 30 2020, the World Health Organization (WHO) recognized SARS-CoV-2 as a public health emergency of international concern and declared it a pandemic on March 11 2020. The virus is transmitted either directly through aerosols, close contact, body fluids and secretions, and mother-to-child, or indirectly via fomites and objects used by the infected person [2]. The virus is highly infectious and can infect peoples of all ages. The average incubation periods last for 5 to 6 days. Fever, dry cough, and fatigue were the main symptoms of COVID-19 [3].

The outbreak of COVID-19 has affected the entire globe and continues to alter life. As of March 24 2021, over 123,902,242 confirmed cases and more than 2,727,837 deaths were reported globally. The United States, Europe, Brazil, and India were primarily affected. In Africa, COVID-19 has caused more than 3,020,998 confirmed cases and 76,598 deaths. In Ethiopia, 190,594 confirmed cases and 2,693 deaths had been reported [4]. As compared to the developed nations, the number of cases and deaths in Africa is low. This might be due to insufficient testing capacity, week contact tracing, and a flawed reporting system. In Ethiopia, the virus continues to spread at an alarming rate. As of March 24 2021, Ethiopia was the third country reporting higher cases of COVID-19 in Africa [4]. The community did not practice the information given by the ministry of health and the government. The political instability and mass demonstrations in the country's capital city and some cities of the Amhara and Oromia region might have led to such crises.

According to Resolve to Save Lives, an initiative of Vital Strategies, the general epidemic preparedness of Ethiopia using the "ReadyScore" criteria is 52%, which indicates that much work is expected from the country [5]. According to a recent study, only 54%, 26.1%, and 53.1% of students had a positive attitude, good preparedness, and high self-efficacy towards COVID-19, respectively. Students are more dynamic and susceptible to acquire the infection and pass to the community. Besides, dorms and student socializing look to be the natural habitats for the virus. Despite this, schools and universities are now reopening with strict coronavirus rules.

During school reopening, Debre Berhan University has specified rules and regulations strictly adhered to by the staff and the students. All staff and students should maintain a 1.5 to 2 meters physical distance from others. In the classroom, students shall not exceed 30 and should be limited to 40 minutes session. All staff and students should wear a face mask. Besides, they are required to frequently clean hands using soap and water or an alcohol-based hand rub for at least 20 seconds. If they feel unwell or have COVID-19 symptoms, they should

stay in their residence and seek medical attention. After the end of each session, classrooms should be ventilated for at least 15 minutes and shall be cleaned and disinfected twice a day. Offices, libraries, laboratories, workshops, conference rooms, and other instructional settings should be cleaned, ventilated, and disinfected regularly for at least an hour a day. Besides, team-building and group activities are prohibited. Furthermore, a committee shall be established to monitor and evaluate the situation in schools in collaboration with the education sector (See S2 File).

However, if recurrence occurs, school administrators should be autonomous or flexible to make rapid and responsive decisions based on the available information, resources, capacities, and needs. Preparations for COVID-19 resurgences should be arranged to support and prepare schools and universities by designing plans and protocols for continuing education and providing training in anticipation of potential recurrences that may need to close, partially close, and reopen more than once. The measures taken for school reopening encourage implementing the existing practices to enable, monitor, and sustain the return to school as equitable and ensuring high-quality education while protecting students' health and well-being [6].

Strict follow-up of precaution measures, i.e., wearing a mask, sanitary measures, keeping physical distance, avoiding crowds, and keeping respiratory hygiene, are the fundamental strategies to control and prevent pandemics. Student's adherence to the precaution measures predominantly relies on their attitude, preparedness, and self-efficacy towards the COVID-19 [7]. These enhance individual self-protection behaviours, willingness to cooperate, and adopt new preventive measures, which positively influences the efforts to prevent infectious transmission. Thus, we intended to assess the attitude, preparedness, self-efficacy, and associated factors of COVID-19 among Debre Berhan University students during campus re-entry, Northeast Ethiopia.

## Method and materials

### Study setting, population, and design

This cross-sectional study was conducted among Debre Berhan University (DBU) students from December 1 to 15/2020, when students just return to campus. The university is found at Debre Berhan town, the administrative centre of North-Shewa Zone, North-East Ethiopia. It is located 130 km far from Addis Ababa, the capital city of Ethiopia. DBU had two institutes, twelve colleges, and 50 departments. By the year 2020/21, there are about 11,573 regular undergraduate and postgraduate students. Of these, 4,419 are females.

All Regular students available at the time of data collection and decided to participate in the study were included. Those who had a physical/mental disability and not competent to fill out the questionnaire were excluded.

### Sample size and sampling procedure

The sample size was determined using Open Epi version 3.03 statistical software. It was calculated for determinant factors and the levels of attitude, preparedness, and self-efficacy to control COVID 19. Then, the maximum sample size was considered with the following assumptions: proportion 65% [8], confidence level 95%, power of the study (1-β), 80%, level of significance, $\alpha$ = 5%, a margin of error, d = 5%, and design effect, D = 2. In the end, a 5% non-response rate was added to give 713.

A multistage sampling technique was applied to select a regular Debre Berhan University student. A simple random sampling technique (SRS) was used to select five representative colleges, and the calculated sample size was proportionally allocated to the selected colleges.

Based on the total number of students, the required sample size was again proportionally allocated to the assigned departments. Then, SRS was done to pick the required sample from a pre-determined sampling frame.

## Study variables

In this study, attitude, preparedness, and self-efficacy towards COVID-19 were the dependent variables. Sociodemographic factors, chronic disease, and friend/family history of COVID-19 infection were considered independent variables.

## Data collection tool and procedures

Data were collected through a self-administered technique using a structured questionnaire. The data collection tool was prepared by referring to previous studies [7, 9–11] and was amended to fit the study population and research objectives. The instrument consisted of five parts: baseline characteristics, chronic illness, and exposure to COVID-19 infection, students' attitudes towards COVID-19 prevention measures, preparedness to combat the spread of COVID-19, and perceived self-efficacy to control and prevent COVID-19.

## Measurement

Ten items were prepared to determine students' attitude towards COVID-19 prevention measures. The items have three alternative options, "Agree", "Neutral", and "Disagree". Two points were provided for those who responded "Agree", one point for "Neutral", and zero points for "Disagree" responses. The mean attitude score was computed, and students who scored greater than or equal to the mean were considered to have a favourable attitude [12]. The Cronbach's alpha coefficients of the items were 0.78.

The level of preparedness to combat the spread of COVID-19 pandemics was assessed using ten (Yes or No) questions. The score ranges from zero to 10 points. Based on the ReadyScore criteria of preparedness created by Resolve to Save lives, students were classified as "Good/Better Prepared" if they scored 80% or higher, "Moderate/Work to do" if scored 40 to 80%, and "Poor/Not Ready" if scored less than 40% [13]. Two dichotomous outcomes indicated the overall level of preparedness: "Good Preparedness" (≥8 points) and "Poor Preparedness" (<8 points). The Cronbach's alpha coefficients of the items were 0.82.

Perceived self-efficacy to control COVID-19 pandemics was assessed using four items that were responded on a three-point Likert scale: "Agree (2 points)", "Neutral (1 point)", and "Disagree (0 points)". Based on the mean score, self-efficacy was categorized as high if scored above the mean and low if scored at the mean or below [7]. The Cronbach's alpha coefficients for the items were 0.69.

## Data quality control

Before the actual data collection, the tool was validated by pre-testing 36 students (5% of the sample) at Victory College, and necessary modification was made. The reliability of the tool was also evaluated using Cronbach's alpha ($\alpha$). Besides, face and content validity were checked whether it appears to measure the construct of interest. A one-day training and orientation regarding the aims, tools, and methods of the study were given for data collectors and supervisors. The supervisors and principal investigators were evaluated the collected data for completeness, consistency, clarity, and missing values.

## Data management and analysis

The data were cleaned, coded, and entered into Epi-data version 4.6 and exported to IBM SPSS statistical software version 25 for analysis. Descriptive analysis was computed, and the findings were presented using texts, frequency tables, means (±SD), and percentages. Attitude, preparedness, and self-efficacy scores were determined to estimate the overall attitude, preparedness, and self-efficacy level. Variables with a p-value of ≤0.25 in the univariable logistic regression analysis were entered into a multivariable logistic regression analysis model. Adjusted odds ratio (AOR) with a 95% confidence interval (CI) was computed to interpret the strength of associations. The Hosmer-Lemeshow goodness-of-fit and Omnibus test were applied to check for model fitness, and a p-value of <0.05 was considered statistically significant.

## Ethical approval

The study was ethically approved by the Institutional Review Board (IRB) of Debre Berhan University, Institute of Medicine and Health Science Review Committee (Ref. No: P0007/20). Participants were informed about the study aims, risks, and benefits, and those who agree to participate provided written informed consent (See S1 File). The study was conducted following the declaration of Helsinki Ethical Principles (1964). Information was kept confidential and anonymous.

# Result

## Background characteristics

Six hundred eighty-two respondents were participated in this study, making a 95.6% response rate. The mean age (+SD) of participants was 23.35±3.46 years and ranged from 18 to 40 years. Participants aged 36 years and older (29.4%) were better prepared to combat the spread of COVID-19. Besides, females (20.1) and rural residents (20.0%) had good preparedness towards COVID-19. One hundred eight (20.2%) single participants had good preparedness, whereas the majority (92.1) who are in an open relationship had poor preparedness ($p = 0.009$). About 21.3% of health students had better preparedness than 11.7% of non-health students ($p = 0.002$) (Table 1).

## Chronic diseases and history of COVID-19 infection

Most heart disease participants (45%) were better prepared to combat COVID-19 infection ($p = 0.000$). Besides, students who had diabetes (25%) and respiratory disease (30%) had good preparedness towards COVID-19. Eighteen (42.9%) of participants who had a friend/ family history of death from COVID-19 infection were more prepared against COVID-19 ($p = 0.000$) (Table 2).

## Attitude towards COVID prevention measures

Six hundred-two (88.3%) of students agreed that frequent hands washing with soap and water prevents COVID-19, and 582(85.3%) agree to wear masks to avoid the infection. Besides, 580 (85%) agreed that suspected peoples' self-quarantine helps prevent COVID-19 (Table 3). The mean score of attitudes was 17.65±32.14, and the minimum and maximum scores were 0 and 20, respectively. The overall proportion of study participants with a positive attitude towards COVID-19 prevention measures was 67.2% (N = 458), 95% CI (63.6–70.5).

## Level of preparedness

About 72.4% of students are ready to wear face masks regularly, 61% frequently clean hands, 39% avoid crowds, and 51.6% coughs and sneezes with a bent elbow or tissue (Table 4). The mean score of preparedness was 4.66±2.96. Overall, 122 (17.9%) of University students had

**Table 1. Distribution of sociodemographic characteristics by level of preparedness in DBU, 2020.**

| Variables | Level of preparedness, n (%) | | *P*-value |
|---|---|---|---|
| | **Poor** | **Good** | |
| **Age** | | | 0.222 |
| 18–25 years | 458(81.6) | 103(18.4) | |
| 26–35 years | 90(86.5) | 14(13.5) | |
| ≥ 36 years | 12(70.6) | 5(29.4) | |
| **Gender** | | | 0.068 |
| Male | 326(84.5) | 60(15.5) | |
| Female | 234(79.1) | 62(20.1) | |
| **Residence** | | | 0.140 |
| Rural | 280(80.0) | 70(20.0) | |
| Urban | 280(84.3) | 52(15.7) | |
| **Religion** | | | 0.144 |
| Christian | 510(81.5) | 116(18.5) | |
| Muslim | 50(89.3) | 6(10.7) | |
| **Marital status** | | | 0.009 |
| Single | 426(79.8) | 108(20.2) | |
| In relationship | 70(92.1) | 6(7.9) | |
| Married | 64(88.9) | 8(11.1) | |
| **Level of education** | | | 0.596 |
| Undergraduate studies | 470(82.5) | 100(17.5) | |
| Postgraduate studies | 90(80.4) | 22(19.6) | |
| **Faculty** | | | 0.002 |
| Health Science | 348(78.7) | 94(21.3) | |
| Non-health Science | 212(88.3) | 28(11.7) | |
| **Mothers' education** | | | 0.268 |
| No formal education | 286(79.4) | 74(20.6) | |
| Primary education | 96(85.7) | 16(14.3) | |
| Secondary education | 86(86.0) | 14(14.0) | |
| Higher education | 92(83.6) | 18(16.4) | |
| **Fathers' education** | | | 0.012 |
| No formal education | 250(82.2) | 54(17.8) | |
| Primary education | 82(73.2) | 30(26.8) | |
| Secondary education | 94(90.4) | 10(9.6) | |
| Higher education | 134(82.7) | 28(17.3) | |
| **Number of students in dorm** | | | 0.319 |
| < 4 | 104(85.2) | 18(14.8) | |
| ≥ 4 | 456(81.4) | 104(18.6) | |

good, 308 (45.1%) moderate, and 252 (37.0%) poor level of preparedness to combat the spread of COVID-19 infection.

## Perceived self-efficacy in controlling COVID-19

About three-fourths (73.3%) of participants believed that they could protect themselves against COVID-19, and 59.8% are confident that Ethiopia can win the battle against COVID-19 (Table 5). The mean cumulative score of self-efficacy was 6.11±1.84. Regarding the comprehensive self-efficacy in COVID-19 prevention, 50.4% (N = 344), 95% CI (46.5–54.1) of the participants had high self-efficacy.

**Table 2. Chronic diseases and history of COVID-19 infection by the level of preparedness in DBU, 2020.**

| Variables | Category | Level of preparedness, n (%) | | P-value |
|---|---|---|---|---|
| | | Poor | Good | |
| **Chronic diseases** | | | | |
| Diabetes | Yes | 24(75.0) | 8(25.0) | 0.282 |
| | No | 536(82.5) | 114(17.5) | |
| Hypertension | Yes | 24(80) | 6(20) | 0.758 |
| | No | 536(82.2) | 116(17.8) | |
| Kidney disease | Yes | 20(76.9) | 6(23.1) | 0.482 |
| | No | 540(82.3) | 116(17.7) | |
| Heart disease | Yes | 22(55.0) | 18(45.0) | 0.000 |
| | No | 538(83.8) | 104(16.2) | |
| Respiratory disease | Yes | 28(70.0) | 12(30.0) | 0.039 |
| | No | 532(82.9) | 110(17.1) | |
| HIV/AIDS | Yes | 18(75.0) | 6(25.0) | 0.355 |
| | No | 542(82.4) | 116(17.6) | |
| **History COVID-19 infection** | | | | |
| Yes | | 154(82.8) | 32(17.2) | 0.775 |
| No | | 406(81.9) | 90(18.1) | |
| **Friends/ families tested positive for COVID-19 infection** | | | | |
| Yes | | 74(75.5) | 24(24.5) | 0.065 |
| No | | 486(83.2) | 98(16.8) | |
| **Friends/ families died from COVID-19 infection** | | | | |
| Yes | | 24(57.1) | 18(42.9) | 0.000 |
| No | | 536(83.8) | 104(16.3) | |

## Predictors of attitude towards COVID-19 prevention measures

In the univariable logistic regression analysis, each variable was computed with the outcome variable; attitude towards COVID-19 prevention measures. Then, variables with a *p*-value less than 0.25 were included in the multivariable logistic regression analysis model.

**Table 3. Students' attitude towards COVID-19 prevention measures in DBU, 2020.**

| Variables | Agree, N (%) | Neutral, N (%) | Disagree, N (%) |
|---|---|---|---|
| Wearing a face mask helps to prevent COVID-19. | 582(85.3) | 78(11.4) | 22(3.2) |
| Avoidance of touching nose, eye, and face with unwashed hand protect COVID-19. | 570(83.6) | 52(7.6) | 60(8.8) |
| Do you think avoiding handshake and kissing with others protect COVID-19? | 550(80.6) | 62(9.1) | 70(10.3) |
| Physical distancing helps to prevent COVID-19. | 568(83.3) | 80(11.7) | 34(5.0) |
| Do you think frequent hands washing with soap and water prevents COVID-19? | 602(88.3) | 66(9.7) | 14(2.1) |
| The use of hand sanitizers helps to prevent COVID-19. | 562(82.4) | 92(13.5) | 28(4.1) |
| Do you think keeping rooms well ventilated helps to prevent COVID-19? | 514(75.4) | 112(16.4) | 56(8.2) |
| Avoiding crowds helps to prevent COVID-19. | 542(79.5) | 88(12.9) | 52(7.6) |
| Do you think coughing into a bent elbow or tissue helps prevent COVID-19? | 538(78.9) | 118(17.3) | 26(3.8) |
| Do you think the self-quarantine of suspected peoples helps to prevent COVID-19? | 580(85.0) | 70(10.3) | 32(4.7) |

**Table 4. Preparedness of students to combat the spread of COVID-19 pandemics in DBU, 2020.**

| Variables | Yes, N (%) | No, N (%) |
|---|---|---|
| Wearing a face mask at all times in public spaces. | 494 (72.4%) | 188 (27.6%) |
| Readiness to avoid touching eye, nose, and face with an unwashed hand. | 296 (43.4%) | 386 (56.6%) |
| Avoid handshakes and kiss with others. | 230 (33.7%) | 452 (66.3%) |
| Maintain physical distance | 230 (33.7%) | 452 (66.3%) |
| I am frequently cleaning hands with soap and water. | 416 (61.0%) | 266 (39.0%) |
| Readiness to use hand sanitizers. | 210 30.8%) | 472 (69.2%) |
| Keeping rooms well ventilated. | 268 (39.3%) | 414 (60.7%) |
| Readiness to avoid crowds. | 266 (39.0%) | 416 (61.0%) |
| Cough into a bent elbow or tissue. | 352 (51.6%) | 330 (48.4%) |
| Readiness to self-quarantine if exposed | 414 (60.7%) | 268 (39.3%) |

In the multivariable logistic regression analysis model, only the mother's education was significantly associated with an attitude towards COVID-19 precaution measures. Students whose mothers had attended secondary education were two times more likely to have a favorable attitude compared to those students whose mothers had no formal education at all (AOR (CI) = 2.05(1.01–4.19) (Table 6).

## Predictors of preparedness to combat the spread of COVID-19

In bivariable logistic regression analysis, 11 variables had p-value < 0.25 and fitted for multivariable regression analysis. Five variables, gender, marital status, faculty, chronic illness, and attitude towards COVID-19 precaution measures, remained predictors of preparedness (Table 7).

Female students are 82% more likely to have good preparedness to combat the spread of COVID-9 infection (AOR (CI) = 1.82(1.16–2.86). Similarly, the odds of good preparedness increased by 70% for students from health science faculty (AOR (CI) = 1.70(1.01–2.85). Respondents who had heart disease were three times more likely to have good COVID-19 preparedness compared to their counterparts (AOR (CI) = 2.91(1.29–6.54). Besides, the favourable attitude was the strongest predictor for preparedness (AOR (CI) = 6.21(3.24–11.9), students who had a favourable attitude were six times more likely to have better preparedness. However, students who are in an open relationship were less prepared compared to those who are single (AOR (CI) = 0.38(0.15–0.95).

## Predictors of perceived self-efficacy in controlling COVID-19

Based on the multivariable logistic regression analysis model, level of education, parents' education, living condition, chronic illness, friends or family history of COVID-19 infection and death, level of attitude, and preparedness were significant factors of perceived self-efficacy towards COVID-19 prevention (Table 8).

**Table 5. Perceived self-efficacy in controlling COVID-19 among DBU students, 2020.**

| Variables | Agree, N (%) | Neutral, N (%) | Disagree, N (%) |
|---|---|---|---|
| I believe I can protect myself against COVID-19. | 500(73.3) | 162(23.8) | 20(2.9) |
| I believe COVID-19 can finally be successfully controlled. | 372(54.5) | 246(36.1) | 64(9.4) |
| I can strictly follow prevention behaviours. | 386(56.6) | 242(35.5) | 54(7.9) |
| I have confidence that Ethiopia can win the battle against COVID-19. | 408(59.8) | 184(27.0) | 90(13.2) |

**Table 6.** Univariable and multivariable logistic regression analysis for attitude towards COVID-19 prevention measures in DBU, Northeast Ethiopia, 2020.

| Variables | Categories | Attitude | | COR (95% CI) | AOR (95% CI) |
|---|---|---|---|---|---|
| | | Unfavorable | Favorable | | |
| **Faculty** | Health | 138(31.2) | 304(68.8) | 1.23(0.88–1.71) | 1.31(0.92–1.87) |
| | Non-health | 86(35.8) | 154(64.2) | 1 | 1 |
| **Mothers' education** | No formal education | 118(32.8) | 242(67.2) | 1 | 1 |
| | Primary education | 46(41.1) | 66(58.9) | 0.70(0.45–1.08) | 0.77(0.45–1.29) |
| | Secondary education | 16(16.0) | 84(84.0) | 2.56(1.44–4.56) | 2.05(1.01–4.19)* |
| | Higher education | 44(40.0) | 66(60.0) | 0.73(0.47–1.14) | 0.59(0.33–1.08) |
| **Fathers' education** | No formal education | 104(34.2) | 200(65.8) | 1 | 1 |
| | Primary education | 44(39.3) | 68(60.7) | 0.80(0.51–1.26) | 0.87(0.53–1.45) |
| | Secondary education | 32(30.8) | 72(69.2) | 1.17(0.73–1.89) | 1.10(0.60–2.02) |
| | Higher education | 44(27.2) | 118(72.8) | 1.39(0.92–2.12) | 1.50(0.83–2.71) |
| **Number of students in dorm** | < 4 | 46(37.7) | 76(62.3) | 1 | 1 |
| | ≥ 4 | 178(31.8) | 382(68.2) | 1.29(0.86–1.95) | 1.19(0.77–1.83) |
| **Having diabetes** | Yes | 14(43.8) | 18(56.3) | 0.61(0.29–1.26) | 0.77(0.33–1.78) |
| | No | 210(32.3) | 440(67.7) | 1 | 1 |
| **Kidney disease** | Yes | 14(53.8) | 12(46.2) | 0.40(0.18–0.89) | 0.54(0.22–1.31) |
| | No | 210(32.0) | 446(68.0) | 1 | 1 |
| **History COVID-19 infection** | Yes | 68(36.6) | 118(63.4) | 0.79(0.56–1.13) | 0.71(0.48–1.05) |
| | No | 156(31.5) | 340(68.5) | 1 | 1 |
| **Friend/family history of COVID-19** | Yes | 22(22.4) | 76(77.6) | 1.83(1.10–3.03) | 1.65(0.93–2.92) |
| | No | 202(34.6) | 382(65.4) | 1 | 1 |
| **Friends/families died from COVID-19** | Yes | 10(23.8) | 32(76.2) | 1.61(0.78–3.33) | 1.71(0.73–3.99) |
| | No | 214(33.4) | 426(66.6) | 1 | 1 |

*Statistically significant at p-value <0.05.

Perceived high self-efficacy was two times higher among undergraduate students (AOR (CI) = 2.12(1.18–3.80), had a favourable attitude (AOR (CI) = 2.57(1.76–3.74), and good preparedness (AOR (CI) = 2.26(1.39–3.66). Similarly, residing in a dorm being four and above had an increased chance for high self-efficacy (AOR (CI) = 1.69(1.06–2.72). Besides, the level of self-efficacy was significantly higher among students who had a friend/family history of COVID-19 infection (AOR (CI) = 1.97(1.14–3.42) and friend/family history of death from COVID-19 (AOR (CI) = 2.82(1.05–7.57).

However, mothers' educational status (higher education) and fathers' educational status (secondary education) decreased the odds of self-efficacy to control COVID-19 by 59% (AOR (CI) = 0.41(0.22–0.76) and 53% (AOR (CI) = 0.47(0.26–0.87), respectively. Respondents who had kidney disease were also less likely to have high self-efficacy (AOR (CI) = 0.32(0.12–0.86).

## Discussion

In this study, the overall level of favourable attitude, good preparedness, and high self-efficacy among students was 67.2%, 17.9%, and 50.4%, respectively. Mothers' education was associated with attitude. Female gender, open relationships, health science faculty, heart disease, and favourable attitude were significant associates of preparedness. Whereas being undergraduate, parents' education, residing in dorm being four and above, having kidney disease, having friend/family history of COVID-19 infection and death, favourable attitude, and good preparedness were predictors of self-efficacy.

**Table 7. Univariable and multivariable logistic regression analysis for preparedness to combat the spread of COVID-19 in DBU, Northeast Ethiopia, 2020.**

| Variables | Categories | Preparedness | | COR (95% CI) | AOR (95% CI) |
|---|---|---|---|---|---|
| | | Poor | Good | | |
| **Gender** | Male | 326(84.5) | 60(15.5) | 1 | 1 |
| | Female | 234(79.1) | 62(20.1) | 1.44(0.97–2.13) | 1.82(1.16–2.86)* |
| **Residence** | Rural | 280(80.0) | 70(20.0) | 1.35(0.91–1.99) | 1.05(0.64–1.71) |
| | Urban | 280(84.3) | 52(15.7) | 1 | 1 |
| **Marital status** | Single | 426(79.8) | 108(20.2) | 1 | 1 |
| | In relationship | 70(92.1) | 6(7.9) | 0.34(0.14–0.79) | 0.38(0.15–0.95)* |
| | Married | 64(88.9) | 8(11.1) | 0.49(0.23–1.06) | 0.44(0.19–1.01) |
| **Faculty** | Health | 348(78.7) | 94(21.3) | 2.05(1.29–3.22) | 1.70(1.01–2.85)* |
| | Non-health | 212(88.3) | 28(11.7) | 1 | 1 |
| **Mothers' education** | No formal education | 286(79.4) | 74(20.6) | 1 | 1 |
| | Primary education | 96(85.7) | 16(14.3) | 0.64(0.36–1.16) | 0.69(0.35–1.38) |
| | Secondary education | 86(86.0) | 14(14.0) | 0.63(0.34–1.17) | 0.75(0.32–1.77) |
| | Higher education | 92(83.6) | 18(16.4) | 0.76(0.43–1.33) | 0.93(0.42–2.07) |
| **Fathers' education** | No formal education | 250(82.2) | 54(17.8) | 1 | 1 |
| | Primary education | 82(73.2) | 30(26.8) | 1.69(1.01–2.82) | 1.82(0.98–3.37) |
| | Secondary education | 94(90.4) | 10(9.6) | 0.49(0.24–1.01) | 0.58(0.24–1.43) |
| | Higher education | 134(82.7) | 28(17.3) | 0.97(0.59–1.59) | 1.12(0.51–2.45) |
| **Heart disease** | Yes | 22(55.0) | 18(45.0) | 4.23(2.19–8.17) | 2.91(1.29–6.54)* |
| | No | 538(83.8) | 104(16.2) | 1 | 1 |
| **Respiratory disease** | Yes | 28(70.0) | 12(30.0) | 2.07(1.02–4.20) | 1.18(0.52–2.73) |
| | No | 532(82.9) | 110(17.1) | 1 | 1 |
| **Friend/family history of COVID-19** | Yes | 74(75.5) | 24(24.5) | 1.61(0.97–2.68) | 1.07(0.53–2.16) |
| | No | 486(83.2) | 98(16.8) | 1 | 1 |
| **Friends/families died from COVID-19** | Yes | 24(57.1) | 18(42.9) | 3.86(2.03–7.38) | 2.28(0.92–5.67) |
| | No | 536(83.8) | 104(16.3) | 1 | 1 |
| **Attitude** | Unfavorable | 212(94.5) | 12(5.4) | 1 | 1 |
| | Favorable | 348(76.0) | 110(24.0) | 5.58(3.0–10.38) | 6.21(3.24–11.9)* |

*Statistically significant at p-value <0.05.

## Attitude towards COVID-19 prevention measures

The study found that 67.2%, 95% CI (63.6–70.5) of university students had a favourable attitude towards COVID-19 precaution measures during school resumption. This finding is almost similar to a study report from Cameroon [14], Pakistan [15], and Japan [16], in which 69%, 65.4%, and 68.5% of students had favourable attitudes, respectively. However, this finding was lower than 73.8% from China [17] and 90.3% from Gedeo Zone [18]. The differences in the study setting might explain this as China has much better resources, health care systems, information, and protection than Ethiopia. Besides, it was done during the lockdown period, in which the fatalities and crises might have increased the attitude of humankind towards COVID-19 prevention. Further, this study used the mean value (88.3%) as a cut point for classifying the attitude level, whereas the Gedeo study used 55.6%.

On the other hand, this finding was higher than the reports from Addis Ababa [19] and Gondar [11], in which 60.7% and 34.1% of students had positive attitudes. The discrepancy might be subjected to variation in measurement tools, level of education, study population, and access to information. These two studies were conducted among communities, including

**Table 8. Univariable and multivariable logistic regression analysis for perceived self-efficacy in controlling COVID-19 pandemics in DBU, Northeast Ethiopia, 2020.**

| Variables | Categories | Self-efficacy | | COR (95% CI) | AOR (95% CI) |
|---|---|---|---|---|---|
| | | **Low** | **High** | | |
| **Age** | 18–25 years | 260(46.3) | 301(53.7) | 1 | 1 |
| | 26–35 years | 68(65.4) | 36(34.6) | 0.46(0.29–0.71) | 0.70(0.41–1.22) |
| | ≥ 36 years | 10(58.8) | 7(41.2) | 0.61(0.23–1.61) | 0.70(0.22–2.21) |
| **Faculty** | Health | 206(46.6) | 236(53.4) | 1.40(1.02–1.92) | 1.23(0.85–1.78) |
| | Non-health | 132(55.0) | 108(45.0) | 1 | 1 |
| **Level of education** | Undergraduate | 266(46.7) | 304(53.3) | 2.06(1.35–3.13) | 2.12(1.18–3.80)* |
| | Postgraduate | 72(64.3) | 40(35.7) | 1 | 1 |
| **Mothers' education** | No formal education | 152(42.2) | 208(57.8) | 1 | 1 |
| | Primary education | 64(57.1) | 48(42.9) | 0.55(0.36–0.84) | 0.95(0.55–1.64) |
| | Secondary education | 44(44.0) | 56(56.0) | 0.93(0.59–1.45) | 1.22(0.64–2.32) |
| | Higher education | 78(70.9) | 32(29.1) | 0.30(0.19–0.48) | 0.41(0.22–0.76)* |
| **Fathers' education** | No formal education | 132(43.4) | 172(56.6) | 1 | 1 |
| | Primary education | 56(50.0) | 56(50.0) | 0.78(0.49–1.18) | 0.61(0.35–1.05) |
| | Secondary education | 60(57.7) | 44(42.3) | 0.56(0.36–0.88) | 0.47(0.26–0.87)* |
| | Higher education | 90(55.6) | 72(44.4) | 0.61(0.42–0.90) | 0.58(0.32–1.07) |
| **Number of students in dorm** | < 4 | 78(63.9) | 44(36.1) | 1 | 1 |
| | ≥ 4 | 260(46.4) | 300(53.6) | 2.04(1.36–3.07) | 1.69(1.06–2.72)* |
| **Kidney disease** | Yes | 18(69.2) | 8(30.8) | 0.42(0.18–0.98) | 0.32(0.12–0.86)* |
| | No | 320(48.8) | 336(51.2) | 1 | 1 |
| **Heart disease** | Yes | 14(35.0) | 26(65.0) | 1.89(0.97–3.69) | 2.10(0.86–5.12) |
| | No | 324(50.5) | 318(49.5) | 1 | 1 |
| **Friend/family history of COVID-19** | Yes | 32(32.7) | 66(67.3) | 2.27(1.44–3.57) | 1.97(1.14–3.42)* |
| | No | 306(52.4) | 278(47.6) | 1 | 1 |
| **Friends/families died from COVID-19** | Yes | 8(19.0) | 34(81.0) | 4.52(2.06–9.93) | 2.82(1.05–7.57)* |
| | No | 330(51.6) | 310(48.4) | 1 | 1 |
| **Attitude** | Unfavorable | 152(67.9) | 72(32.1) | 1 | 1 |
| | Favorable | 186(40.6) | 272(59.4) | 3.08(2.21–4.32) | 2.57(1.76–3.74)* |
| **Preparedness** | Poor | 304(54.3) | 256(45.7) | 1 | 1 |
| | Good | 34(27.9) | 88(72.1) | 3.07(2.00–4.72) | 2.26(1.39–3.66)* |

*Statistically significant at P-value <0.05.

traditional healers and religious clerics. These groups are older, had no formal education, and had poor access to information which could affect the level of knowledge and attitude towards the pandemics.

Students whose mothers had attended secondary education were more likely to have a favourable attitude compared to those whose mothers had no formal education at all. Similarly, educational status was significantly related to a positive attitude in Gedeo Zone [18]. Besides, studies conducted in South Wollo [8] and Mizan Tepi [20] reported better attitudes among educated participants. When the education level increases, the students' attitude and knowledge towards the prevention measures will also increase [11].

## Preparedness to combat the spread of COVID-19 pandemics

According to Resolve to Save Lives, an initiative of Vital Strategies, the general epidemic preparedness of Ethiopia using the "ReadyScore" criteria is 52%, which indicates that much work

is expected from the country [5]. In this study, students in an overall 17.9%, 95% CI (15.0–21.0) had good preparedness to avoid COVID-19. A similar study in Ghana indicated that only 14% of participants had good preparedness skills to prevent and control COVID-19 infection [9]. Students are probably careless and reluctant because of fewer deaths confirmed in Ethiopia. However, this outcome was lower than 26.1% in South Wollo [21], 54% in North Shoa [22], and 41.3% in Gondar [10]. The variation may account for the differences in the target population, study period, outcome measurement, and sample size. These studies were done among frontline health care providers, health institutions, and chronic patients who were supposed to have better preparedness. If the educated sector of Ethiopia has poor preparedness, one may wonder what the preparedness of the general public will be.

Females showed significantly higher levels of preparedness than males, which can decrease the risk of infection. Women's higher perception and better compliance regarding COVID-19 related preparedness were also found in other studies [10, 23]. Besides, females were more likely to have better knowledge, a favourable attitude, and good practice behaviours [24]. This could be due to females spent most of their time at home and naturally more prone to practice preventive measures, such as hand washing and keeping physical distance. Additionally, twice as many men have been dying from COVID-19 as women in the US. Similarly, 69% of all COVID-19deaths across Western Europe occurred among men. This might be related to gender-based lifestyle choices and behavioral differences, for instance in smoking which affect the level of pre-existing disease such as chronic lung disease and heart disease [25].

Students from the health science faculty showed a higher score for COVID-19 related preparedness. Similarly, according to a study by [26], medical students had good pandemic preparedness. Besides, in Egypt, medical students perceive more stress than non-medical [27]. This might be because health students engage in practical attachment that involves contact with patients. Additionally, there is an assumption that these students are an available resource during a crisis. These all make health students vulnerable to contract the infection and hence are more likely to have good preparedness and response to COVID-19.

Students in an open relationship were less prepared to combat the spread of COVID-19. This was represented in South Gondar, where unmarried participants increased the odds of low preparedness [10]. This could be due to marital and romantic relationships expanding a sense of support and caring among friends and families.

Respondents who had heart disease were three times more likely to have higher preparedness and response to COVID-19 infection prevention and control. This was comparable with the study reported in North Shoa, which revealed that chronic patients, i.e., hypertension, diabetes, and tuberculosis, were better prepared to prevent COVID-19 [22]. Chronic patients are linked with more severe COVID-19 infection and mortality and had poorer treatment outcomes, particularly tuberculosis. Because they recognize their illness and susceptibility, they are likely to be prepared.

A favourable attitude towards COVID-19 was found to influence preparedness skills positively. This has been supported by another study in Bangladesh [23]. There is also a significant positive association between attitude and practice of precaution measures [17, 20]. Positive attitudes possibly increase the risk perception towards the pandemics, which pushes participants to better preparedness and practice preventive behaviours.

## Self-efficacy in controlling COVID-19

About three-fourths (73.3%) of participants believed that they could protect themselves against COVID-19, while 56.6% had confidence that they can strictly follow prevention measures. Overall, the comprehensive high self-efficacy in COVID-19 prevention was 50.4%, 95% CI

(46.5–54.1). This finding is consistent with the study reporting that 53.1% of study participants had high self-efficacy towards COVID-19 prevention measures in Southwest Ethiopia [7]. Besides, a study across five continents demonstrated a significant drop in self-efficacy beliefs from before to during the lockdown [28]. Because self-efficacy significantly predicts adherence to precaution measures, students need to ensure self-efficacy to prevent COVID-19 [7].

Our finding revealed that parent's education significantly decreased the level of self-efficacy to control and prevent COVID-19. Respondents who had kidney disease were also less likely to have high self-efficacy. The possible explanation might be that a more significant number of parents had no formal education, had poor access to social media, and are not getting updated information regarding the prevention methods and effective vaccines. Similarly, this study found that undergraduate students had high self-efficacy than postgraduate students in controlling COVID-19. This could be because most postgraduate students are engaged in marital relationships having many family members. Thus, they may feel that one can acquire and spread the virus to the families. Additionally, this study found that the odds of self-efficacy were significantly higher among students who had a friend/family history of COVID-19 infection and death. Having a friend/family history of COVID-19 may alert students to cautiously get prepared and practice protective behaviours that increase the self-efficacy towards preventing the pandemics.

Students who reside in a dorm being four and above had high self-efficacy. This might have opened a chance for discussion, sharing up-to-date information, and positive criticisms among colleagues that may encourage self-efficacy beliefs. It was also observed that the higher the attitude and preparedness towards COVID-19, the higher the perceived self-efficacy would be. This is similar to the study finding reported in North Shoa Zone that showed a statistically significant association between good preparedness and preventive behaviour and high perceived self-efficacy [7]. Self-efficacy beliefs are the foundation for motivation, well-being, and personal accomplishment. Thus, empowering people to enhance their self-efficacy could bring the desired level of preparedness and behavioural changes.

## Conclusion and recommendation

The level of attitude, preparedness, and self-efficacy towards COVID-19 among students during campus re-entry were low. Sociodemographic factors, chronic illness, family/friend history of COVID-19 infection, and death were significant predictors of attitude, preparedness, and self-efficacy towards COVID-19. Managing chronic diseases and raising the attitude and preparedness of students is essential to reduce the burden of COVID-19 pandemics. Besides, emphasis should be placed on male, unmarried, postgraduate, and non-health science faculty students to increase the level of preparedness and self-efficacy. Since university students live in a school compound and have an insufficient socioeconomic basis which makes them susceptible to contract and spread COVID-19, they should have to develop a positive attitude, get prepared and ensure self-efficacy to prevent and control the COVID-19 pandemics.

## Limitation

These study findings should be interpreted with the following limitations. Since self-reported data were used, students might provide socially acceptable answers, resulting in social desirability bias. This is a cross-sectional study and cannot verify the cause-and-effect relationships. Limited studies are available, especially for preparedness and self-efficacy, making our discussion somewhat shallow. Further, the study was conducted only in one institution which might limit its external validity. Longitudinal studies are recommended further to explore the independent predictors and more representative reports. Despite these limitations, the study

provides important information regarding the attitude, preparedness, and self-efficacy to prevent COVID-19 infections during campus re-entry.

## Supporting information

**S1 File. Questionnaire.**
(DOCX)

**S2 File. Academic rules and regulations for Covid-19.**
(PDF)

**S1 Dataset.**
(SAV)

## Acknowledgments

We would like to thank Debre Berhan University for the approval of ethical clearance. I also want to extend my appreciation to the data collectors, supervisors, and study participants for giving their truthful information.

## Author Contributions

**Conceptualization:** Mesfin Tadese.

**Data curation:** Mesfin Tadese.

**Formal analysis:** Mesfin Tadese.

**Methodology:** Mesfin Tadese.

**Software:** Mesfin Tadese.

**Supervision:** Abebe Mihretie.

**Writing – original draft:** Mesfin Tadese.

**Writing – review & editing:** Abebe Mihretie.

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
