## [Decision Letter · Decision Letter 0]

12 Jul 2021

Attitude, preparedness, and perceived self-efficacy in controlling COVID-19 pandemics and associated factors among university students during school reopening

PONE-D-21-11126

Dear Dr. Tadese,

We’re pleased to inform you that your manuscript has been judged scientifically suitable for publication and will be formally accepted for publication once it meets all outstanding technical requirements.

Kind regards,

Jianguo Wang, PhD

Academic Editor

PLOS ONE

1. Please include additional information regarding the survey or questionnaire used in the study and ensure that you have provided sufficient details that others could replicate the analyses. For instance, if you developed a questionnaire as part of this study and it is not under a copyright more restrictive than CC-BY, please include a copy, in both the original language and English, as Supporting Information.

3. Thank you for including your ethics statement:  "N/A".   

Please provide additional details regarding participant consent. In the ethics statement in the Methods and online submission information, please ensure that you have specified what type you obtained (for instance, written or verbal, and if verbal, how it was documented and witnessed). If your study included minors, state whether you obtained consent from parents or guardians. If the need for consent was waived by the ethics committee, please include this information.

Reviewers' comments:

Reviewer's Responses to Questions

**Comments to the Author**

1. Is the manuscript technically sound, and do the data support the conclusions?

Reviewer #1: Yes

2. Has the statistical analysis been performed appropriately and rigorously? 

Reviewer #1: Yes

3. Have the authors made all data underlying the findings in their manuscript fully available?

Reviewer #1: Yes

4. Is the manuscript presented in an intelligible fashion and written in standard English?

Reviewer #1: Yes

5. Review Comments to the Author

Reviewer #1: Firsltly let me thank the authors for the great work they have provided, the work is excellent and is has been an enjoyable experiencing reading it. The topic is of high interest which they have brought, however they may consider looking unto these few remarks. All the best.

Introduction

The authors may consider talking about second wave of Covid-19. If the authors may provide an evidence of the rules laid down by the university as citation of the memo or any document where these rules are laid will add value to the paper.

Methodology

What was the criteria for scoring like 80%above is better/good, the authors may decide to provide justification?

Data analysis

Well written

Results

The authors should consider rearranging their results presentation to suite with the topic; (Attitude, preparedness, and perceived self-efficacy in controlling COVID-19 pandemics

and associated factors among university students during school reopening) for consistency sake, they should start with attitude then ……

Table 1, kindly they should indicate the level of p-values by putting star(*).

Table 6,7 and 8: below the table, the authors may consider indicating that….. NB: Bold figures represent/mean………

Discussion

Well written

Conclusion

Well written

Limitation

The authors may also consider highlighting a limitation of their findings as they only based on one University in Ethiopia.

References

Well Updated

6. PLOS authors have the option to publish the peer review history of their article (what does this mean?). If published, this will include your full peer review and any attached files.

Reviewer #1: **Yes: **George N. Chidimbah Munthali

---

## [Editor Report · Acceptance letter]

26 Aug 2021

PONE-D-21-11126 

Attitude, preparedness, and perceived self-efficacy in controlling COVID-19 pandemics and associated factors among university students during school reopening 

Dear Dr. Tadese:

I'm pleased to inform you that your manuscript has been deemed suitable for publication in PLOS ONE. Congratulations! Your manuscript is now with our production department. 

Kind regards, 

on behalf of

Dr. Jianguo Wang 

Academic Editor

PLOS ONE